

**Liquid-liquid phase separation in secondary organic aerosol particles**
**produced from α-pinene ozonolysis and α-pinene photo-oxidation**
**with/without ammonia**
Suhan Ham[1], Zaeem Bin Babar[2], Jaebong Lee[3], Hojin Lim[2], Mijung Song[1*]
[1] {Department of Earth and Environmental Sciences, Chonbuk National University, Jeonju,
Jeollabuk-do, Republic of Korea}
[2] {Department of Environmental Engineering, Kyungpook National University, Daegu,
Repubilc of Korea}
[3] {Thermal Hydraulics & Severe Accident Research Division, Korea Atomic Energy
Research Institute, Daejeon, Repubilc of Korea}
Correspondence to: Mijung Song (mijung.song@jbnu.ac.kr)
**Abstract**
Recently, liquid–liquid phase separation (LLPS) of secondary organic aerosol (SOA) particles
free of inorganic salts has been intensively studied because of their importance on cloud
condensation nuclei (CCN) properties. Herein, we investigated LLPS in four different types of
SOA particles generated from α-pinene ozonolysis and α-pinene photo-oxidation in the absence
and presence of $NH_3$. LLPS was observed in SOA particles produced from α-pinene ozonolysis
at ~95.8% relative humidity (RH) and α-pinene ozonolysis with $NH_3$ at ~95.4% RH. However,
LLPS was not observed in SOA particles produced from α-pinene photo-oxidation and α-
pinene photo-oxidation with $NH_3$. With datasets of average oxygen to carbon elemental ratio
(O:C) for different types of SOA particles of this study and previous studies, LLPS occurred
when the O:C ratio was less than ~0.44 and LLPS did not occur when the O:C ratio was greater
than ~0.40. When LLPS was observed, the two liquid phases were present up to ~100% RH.
This result can help to predict more accurate results of CCN properties of organic aerosol
particles.



## 1 Introduction

Secondary organic (SOA) particles in the atmosphere can be formed by the oxidation of volatile organic compounds (VOCs) emitted from biogenic and anthropogenic sources (Hallquist et al., 2009). These SOA particles comprise ~20–80% of ultrafine aerosol particles depending on the location (Zhang et al., 2007; Jimenez et al., 2009). They can affect the energy balance of the Earth by scattering and absorbing solar radiation and also by acting as nuclei for cloud formation (Kanakidou et al., 2005; Hallquist et al., 2009; IPCC, 2013; Knopf et al., 2018). In addition, these particles can affect air quality and human health (Kanakidou et al., 2005; Jang et al., 2006; Solomon et al., 2007; Baltensperger et al., 2008; Murray et al., 2010; Wang et al., 2012; Poschl and Shiraiwa, 2015; Shiraiwa et al., 2017).

Many previous studies showed that SOA particles can be formed more efficiently in the presence of gaseous species such as ammonia ($NH_3$) (Zhang et al., 2004; Na et al., 2006; Na et al., 2007; Laskin et al., 2014; Liu et al., 2015a; Liu et al., 2015b; Babar et al., 2017). $NH_3$ is one of the abundant and reactive gaseous species in the atmosphere (Reis et al., 2009; Heald et al., 2012; Reche et al., 2015; Zheng et al., 2015; Sharma et al., 2016; Warner et al., 2016). The chemical composition of SOA particle can be influenced by the reaction with $NH_3$ (Laskin et al., 2015; Liu et al., 2015b), but it is still poorly understood.

Aerosol particles containing SOAs can undergo phase transitions in the atmosphere as relative humidity (RH) changes. So far, many researchers have focused on phase transitions, especially liquid–liquid phase separation (LLPS) in particles containing SOAs and inorganic salts during changes to RH (Pankow et al., 2003; Marcolli et al., 2006; Ciobanu et al., 2009; Bertram et al., 2011; Krieger et al., 2012; Song et al., 2012a; Song et al., 2012b; Zuend and Seinfeld., 2012; Veghte et al., 2013; O'Brien et al., 2015). They established that LLPS always occurred in SOA particles mixed with inorganic salts when the oxygen to carbon elemental ratio (O:C) of the organic materials is smaller than 0.56, while LLPS never occurred when the O:C of the organic materials is greater than 0.80. LLPS commonly occurred in the intermediate O:C ratio range (Bertram et al., 2011; Krieger et al., 2012; Song et al., 2012a; Song et al., 2013; You et al., 2013; You et al., 2014). LLPS in a mixture of SOA particles and inorganic salts is known to affect optical properties (Fard et al., 2018), gas–particle partitioning (Zuend et al., 2010; Zuend and Seinfeld, 2012; Shiraiwa et al., 2013), reactivity (Kuwata and Martin., 2012), hygroscopic



properties (Hodas et al., 2016), and cloud condensation nuclei (CCN) properties of these
particles (Ovadnevaite et al., 2017).
More recently, researchers have focused on LLPS in SOA particles in the absence of inorganic
salts (Peters et al., 2006; Renbaum-Wolff et al., 2016; Rastak et al., 2017; Song et al., 2017;
Song et al., 2018) since it is important to explore the CCN properties of the particles (Petters
et al., 2006; Hodas et al., 2016; Renbaum-Wolff et al., 2016; Ovadnevaite et al., 2017; Rastak
et al., 2017; Liu et al., 2018). Renbaum-Wolff et al. (2016) and Song et al. (2017) observed
LLPS at a high RH of ~95–100% in SOA particles produced from ozonolysis of α-pinene, β-
caryophyllene, and limonene. However, Rastak et al. (2017) and Song et al. (2017) did not
observe LLPS in SOA particles produced from photo-oxidation of isoprene and toluene. The
occurrence of LLPS in SOA particles free of inorganic salts was related to the average O:C of
the organic materials. When the average O:C of the SOA particle is less than ~0.44, LLPS was
observed in the SOA particles free of inorganic salts (Renbaum-Wolff et al., 2016; Rastak et
al., 2017; Song et al., 2017). Song et al. (2018) studied organic particles consisting of one and
two commercially available organic species free of inorganic salts and found that the average
O:C of the organic material can be an important parameter to predict LLPS. LLPS was observed
in particles containing one organic species at an O:C ratio of $\leq 0.44$ and in particles containing
two organic species at an O:C ratio of $\leq 0.58$. However, additional information is needed to
explore LLPS in organic particles relevant to the atmosphere.
Herein, we investigated LLPS in SOA particles produced from ozonolysis and photo-oxidation
of α-pinene. Moreover, we studied the effects of $NH_3$ on SOA particles produced from
ozonolysis and photo-oxidation of α-pinene on the occurrence of LLPS.
**2 Experimental**
**2.1 Production of SOA particles**
Four different types of SOA particles were generated in the flow tube reactor of Kyungpook
National University (KNU), Korea: those produced via α-pinene ozonolysis and α-pinene
photo-oxidation in the absence of $NH_3$ (Table 1), and those produced via α-pinene ozonolysis
and α-pinene photo-oxidation in the presence of $NH_3$ (Table 2). The method of SOA particle



generation was described previously by Babar et al. (2017). The flow tube reactor was run at a
flow rate of 4.0 L·min$^{-1}$, with a residence time of 3.625 min at ~10% RH.
α-pinene of 1000 ppb concentration was injected into the flow tube reactor to produce SOA
particles via ozonolysis without NH$_3$. O$_3$ was produced by passing high purity O$_2$ through a
UV lamp (λ = 185 nm) and was injected into the flow tube reactor at a concentration of 10000
ppb. Table 1 presents the experimental conditions for the ozonolysis.
To produce SOA particles via photo-oxidation in the absence of NH$_3$, 1000 ppb of α-pinene
was injected in the flow tube reactor (Table 1). OH radical was produced by photo-dissociation
of O$_3$ by irradiating O$_3$ with UV (λ = 254 nm) in the presence of water vapor. The following
photochemical reactions take place.
$$O_3 + hv \; \rightarrow \; O_2 + O \tag{1}$$
$$O + \; H_2O \; \rightarrow 2OH \tag{2}$$
Assuming an atmospheric OH concentration of $1.5 \times 10^6$ molecules·cm$^{-3}$, OH exposures were
$8.2 \times 10^{10}$ molecules·cm$^{-3}$·s and $2.3 \times 10^{11}$ molecules·cm$^{-3}$·s, corresponding to atmospheric
aging time of 0.5 d and 2.5 d, respectively, and the concentrations of O$_3$ in the reactor were
2000 and 6000 ppb at 10% RH, respectively.
The same method was used for SOA particle generation via ozonolysis and photo-oxidation in
the presence of NH$_3$, the exception being that NH$_3$ was injected into the flow tube reactor
during particle generation. The concentration of NH$_3$ was 2000 ppb for the ozonolysis and
photo-oxidation (Table 2).
The 4 L·min$^{-1}$ mainstream flow of SOA particles at the outlet of the flow tube reactor was
diluted by a humidified air stream (RH of 60%) of 7 L·min$^{-1}$. After dilution, the mass
concentrations of the SOA particles were measured to range between ~480 μg·m$^{-3}$ and 670
μg·m$^{-3}$ for the four types of SOA particles using a Sequential Mobility Particle Sizer (SMPS+C,
Grimm, Germany). The sample and sheath flow rates of the SMPS were 0.3 and 3.0 L·min$^{-1}$,
respectively. The SOA particles were collected at the outlet of the reactor on a siliconized
substrate (siliconized glass slides of 18 mm, Hampton Research, USA) of size 1–5 μm.



For each experiment, the siliconized glass slide was initially cleaned three times with water
and methanol. Then, it was dried by purging $N_2$ gas. Finally, it was fixed in the Stage D collector
plate of a Sioutas cascade impactor (225-370, SKC, USA), operated at 9 L·min⁻¹.
**2.2 Observation of liquid-liquid phase separation in SOA particles**
The observation of LLPS in a particle required particle diameters of 20–80 μm. In order to
obtain the appropriate particle sizes for the LLPS experiments, SOA particles sized 1–5 μm
collected on the siliconized substrate from the flow tube reactor were placed into a RH-
controlled flow-cell coupled to an optical microscope (Olympus BX43, 40× objective) (Parsons
et al., 2004; Pant et al., 2006; Bertram et al., 2011; Song et al., 2012b; Song et al., 2018) at
~100% RH, and then, the particles grew and coagulated for ~60 min. This process resulted in
a particle size of 20–80 μm (Renbaum-Wolff et al., 2016). Once the particle size was
appropriate for the LLPS experiments, humidity cycles were performed.
During a humidity cycle, RH was reduced from ~100% to ~0%, and then, it was increased to
~100% at a rate of 0.5–1.0% RH·min⁻¹ if LLPS was not observed. If LLPS was observed, the
RH was reduced from ~100% to ~5–10% lower than the RH at which the two liquid phases
merged into one phase, followed by an increase to ~100% RH at a rate of 0.1–0.5% RH·min⁻¹.
The optical images of the SOA particles during the experiment were recorded every 5 s using
a complementary metal oxide semiconductor detector (Digiretina 16, Tucsen, China). All the
experiments were performed at a temperature of 289±0.2 K.
The RH was controlled by the ratio of $N_2/H_2O$ gas at a total flow rate of 500 sccm. The RH
inside the flow-cell was determined using a temperature and humidity sensor (Sensirion SHT
71, Switzerland) which was calibrated by observing the deliquescence RH for the following
pure inorganic salts at 293 K: potassium carbonate (44% RH), sodium chloride (76% RH),
ammonium sulfate (80.5% RH), and potassium nitrate particle (93.5% RH) (Winston and Bates,
1960). The uncertainty of the RH after calibration was ±2.0%.
**3 Results and Discussion**
**3.1 SOA particles produced from α-pinene ozonolysis and α-pinene photo-oxidation**
SOA particles generated by α-pinene ozonolysis with a mass concentration of 500 μg·m⁻³





underwent humidity cycles at 289 ± 0.2 K. Figure 1 shows examples of optical images of a
SOA particle (α-pinene $O_3$ #1 in Table 1) produced from α-pinene ozonolysis with increasing
RH. Only one phase was observed from 0 to ~96% RH (Fig. 1). At 96.6% RH, LLPS occurred
by a mechanism of spinodal decomposition, which distributes many small inclusions (Schlieren)
throughout a particle (Ciobanu et al., 2009; Song et al., 2012b). After phase separation, at ~97.0%
RH, small droplets grew and coagulated to form inner and outer phases in the particle. As the
RH increased further, the SOA particle displayed a core–shell morphology consisting of inner
and outer phases. The two liquid phases co-existed up to ~100% RH, as shown in Fig. 1. When
the RH decreased from ~100%, the inner phase became smaller and merged into one phase at
~95.0% RH. We assume that the inner phase is a water-rich phase and the outer phase is an
organic-rich phase since the size of the inner phase depends on changes to RH (Renbaum-Wolff
et al., 2016; Song et al., 2017). Moreover, previous studies using surface tension, spreading
coefficient, Raman spectroscopy, atomic force microscopy, and scanning electron microscopy
showed consistent results with regard to the morphology of the particles (Jasper, 1972;
Kwamena et al., 2010; Reid et al., 2011; Song et al., 2013; O'Brien et al., 2015; Gorkowski et
al., 2016; Gorkowski et al., 2017; Zhang et al. 2018).
Table 1 summarizes the separation relative humidity (SRH) upon moistening and the merging
relative humidity (MRH) upon drying. In all cases, the SOA mass concentration was ~500
µg·m$^{-3}$. LLPS was observed at 95.8 ± 2.3% RH for all SOA particles derived from α-pinene
ozonolysis, and the two phases merged into one phase at 92.9 ± 4.6% RH.
Renbaum-Wolff et al. (2016) observed LLPS in SOA particles derived from α-pinene
ozonolysis at ~95% RH. It is consistent with our result. They also showed that LLPS in the
particles did not depend on SOA particle mass concentrations between 75 and 11000 µg·m$^{-3}$.
Since the SOA particle mass concentration does not affect LLPS, in this study, we only focused
on the SOA particle mass concentration of ~500 µg·m$^{-3}$ for different types of SOA particles.
We also performed humidity cycles for SOA particles of mass concentration ~500 µg·m$^{-3}$
derived from α-pinene photo-oxidation. Table 1 summarizes the results of the humidity cycles.
None of the SOA particles from α-pinene photo-oxidation underwent LLPS during the RH
cycles. Figure 2 shows examples of optical images of a SOA particle (α-pinene OH #2 in Table
1) for increasing RH. From 0 to 100% RH, there was no evidence of occurrence of LLPS in





the particles.
**3.2 SOA particles produced from α-pinene ozonolysis with NH₃ and α-pinene photo-**
**oxidation with NH₃**
Ammonia is an abundant and reactive gaseous species in the atmosphere (Reis et al., 2009;
Heald et al., 2012; Reche et al., 2015; Zheng et al., 2015; Sharma et al., 2016; Warner et al.,
2016). Previous studies showed that in the presence of NH₃, SOA particles can be formed more
effectively (Zhang et al., 2004; Na et al., 2006; Na et al., 2007; Liu et al., 2015a; Liu et al.,
2015b; Babar et al., 2017). To investigate the effect of NH₃ on LLPS in SOA particles, we
studied LLPS in SOA particles using α-pinene ozonolysis and photo-oxidation in the presence
of NH₃. Table 2 presents the experimental conditions for the particle generation. We used the
experimental conditions of SOA particle generation via ozonolysis and photo-oxidation (Table
1) in this case too, but we injected 2000 ppb of NH₃ into the flow tube reactor during particle
generation (Table 2).
We performed humidity cycles for the SOA particles produced from α-pinene ozonolysis in the
presence of NH₃ for the mass concentration of 500 μg·m⁻³. Figure 3 shows examples of the
optical images of SOA particles produced by α-pinene ozonolysis in the presence of NH₃ as a
function of increasing RH (α-pinene O₃/NH₃ #1 in Table 2). Upon moistening, only one phase
was present (Fig. 3). As RH increased, the one phase of the SOA particle was separated into
two phases at 95.3% RH, the underlying mechanism being spinodal decomposition. At 95.6%
RH, small inclusions in the particle coagulated and grew, and then, as RH increased further, a
core–shell morphology, with a shell consisting of an organic-rich phase and the core consisting
of a water-rich phase on a substrate, were observed. The two liquid phases co-existed up to
~100% RH. When the RH decreased from ~100% RH, the inner phase of the particle became
smaller, and eventually, the inner phase merged into one phase at 94.4% RH.
Table 2 summarizes the results of average SRH and MRH during the humidity cycles for the
SOA particles produced by α-pinene ozonolysis in the presence of NH₃. LLPS occurred at 95.4
± 2.9% RH, and the two phases merged into one phase at 94.4 ± 2.7% RH for the all particles
(Table 2).
For SOA particles derived from α-pinene photo-oxidation in the presence of NH₃, no LLPS



was observed during changes to RH. Table 2 lists the results of SRH and MRH for two different
SOA particles derived from α-pinene photo-oxidation in the presence of $NH_3$. Figure 4 show
the examples of the optical images of SOA particles produced by α-pinene with $NH_3$ photo-
oxidation for increasing RH (α-pinene OH/$NH_3$ #2 in Table 2). Only one phase was observed
from 0 to 100% RH.
**3.3 Phases of the four different types of SOA particles**
Figure 5 shows the RH at which two liquid phases were observed during RH scanning for the
four different types of SOA particles. Circles represent MRH upon drying, and triangles
represent SRH upon moistening. In the figure, the values of SRH and MRH of SOA particles
derived from α-pinene ozonolysis by Renbaum-Wolff et al. (2016) are also included (in red).
If RH equals 0%, no LLPS was observed.
Among the four different types of SOA particles, two types of particles underwent LLPS but
the remaining particles did not (Fig. 5). For the SOA particles derived from α-pinene ozonolysis,
two liquid phases existed at ~95.8 ± 2.3% RH up to ~100 ± 2.0% RH with increasing RH. For
values lower than ~92.9 ± 4.6% RH with decreasing RH, only one phase was observed. For the
SOA particles derived from α-pinene ozonolysis in the presence of $NH_3$, the RH range for the
two liquid phases was ~95.4 ± 2.9% and ~100 ± 2.0% with increasing RH. SRH values of both
SOA particles were very similar within the uncertainties of the measurements. Also, Fig. 5
showed that the values of SRH upon moistening and MRH upon drying for the two types of
particles were close within the uncertainties of the measurements, suggesting that the kinetic
barrier to LLPS in the particles is low. Compared to the SOA particles derived from α-pinene
ozonolysis and from α-pinene ozonolysis with $NH_3$, LLPS was not observed in SOA particles
derived from α-pinene photo-oxidation without/with $NH_3$ (Fig. 5). In these cases, only one
phase was present between 0 and 100% RH.
**3.4 Relation between O:C ratio and LLPS**
Recent studies have shown that occurrence of LLPS in SOA particles free of inorganic salts is
related to the average O:C ratio of the organic materials (Renbaum-Wolff et al., 2016; Rastak
et al., 2017; Song et al., 2017). They showed that LLPS can occur in SOA particles derived
from α-pinene, limonene, and β-caryophyllene for RH between ~95% and ~100% when the




average O:C ratio ranged from 0.34 and 0.44. LLPS was not observed in SOA particles derived
from isoprene and toluene when the average O:C ratio was between 0.52 and 1.30. Figure 6
illustrates LLPS as a function of the average O:C ratio of SOA particles from previous studies
(Lambe et al., 2015; Li et al., 2015; Renbaum-Wolff et al., 2016; Rastak et al., 2017; Song et
al., 2017). Also presented in Table S1 are the average O:C ratios and experimental conditions
for the SOA particles produced without NH₃. In this study, data on the average O:C ratios were
not available, and thus, we chose the O:C ratios in the literature that were closest to the
experimental conditions (Table S1). The O:C ratio for the SOA particles derived from α-pinene
ozonolysis ranges from 0.42–0.44 as per Li et al. (2015), whereas that for SOA particles derived
from α-pinene photo-oxidation is 0.40–0.90 according to Lambe et al. (2015).
According to the dataset of average O:C ratios of different types of SOA particles from this
study as well as previous studies, Fig. 6 shows that LLPS occurred when the average O:C ratio
was between 0.34 and 0.44. This range of the O:C ratio required for occurrence of LLPS in the
SOA particles is consistent with that of previous work (Renbaum-Wolff et al., 2016; Rastak et
al., 2017; Song et al., 2017). However, LLPS did not occur when the average O:C ratio was
between 0.40 and 1.30. The range of O:C ratio corresponding to the absence of LLPS is wider
than that reported by a previous work (Song et al., 2017). The difference could be attributed to
the fact that the SOA particles were generated from different types of VOCs.
Previous studies found nitrogen-containing SOA species in the presence of NH₃ (Laskin et al.,
2015; Liu et al., 2015b). They suggested that ammonium carboxylates were formed by
neutralization between carboxylic acid and ammonia, and amines were formed by carbonyl and
ammonia via Schiff's base reaction (Na et al., 2006; Na et al., 2007; Laskin et al., 2015). The
nitrogen to carbon (N:C) ratio was reported to be 0.01–0.08 based on aerosol mass
spectrometry (AMS) and fourier transform ion cyclotron resonance (FT-ICR MS) (Laskin et
al., 2014; Liu et al., 2015b). It is noteworthy that ammonium carboxylates and amines are
highly water soluble compounds. However, more accurate data for O:C ratios of the SOA
particles in the presence of NH₃ is needed.
Figure 6 also showed the range of the two liquid phases. The two phases consisting of an
organic-rich shell were observed at RH as high as ~100% in all cases. This result can be
important for the CCN properties of organic particles (Petters et al. 2006; Hodas et al. 2016;




Renbaum-Wolff et al., 2016; Ovadnevaite et al., 2017; Rastak et al., 2017; Liu et al., 2018).
LLPS can give an additional insight into attempting more accurate predictions of the CCN
properties of organic particles.

**4 Summary**

We conducted humidity cycles at a temperature of $289 \pm 0.2$ K for four different SOA particles
derived from α-pinene ozonolysis, α-pinene photo-oxidation, α-pinene ozonolysis with $NH_3$,
and α-pinene photo-oxidation with $NH_3$, for particle mass concentrations of ~500 μg·m$^{-3}$.
Among the four different types of SOA particles, LLPS occurred in SOA particles produced
from α-pinene ozonolysis at $95.8 \pm 2.3\%$ RH with increasing RH and in those produced from
α-pinene ozonolysis with $NH_3$ at $95.4 \pm 2.9\%$ RH with increasing RH. In both types of particles,
the two liquid phases co-existed up to ~100% RH. However, LLPS was not observed in SOA
particles produced from α-pinene photo-oxidation and α-pinene photo-oxidation with $NH_3$.
Analysis of the dataset of average O:C ratios of different types of SOA particles from this study
and previous studies indicated that LLPS occurred when the O:C ratio was less than ~0.44, and
LLPS did not occur when the O:C ratio was greater than ~0.40.
Considering the range of the O:C ratio of organic particles in the atmosphere (0.2–1.0), these
results provide additional evidence that LLPS can occur in organic particles even without the
presence of inorganic salts in the atmosphere. Moreover, LLPS occurred in the SOA particles
at high RH (as high as ~100%), implying that these results can provide additional information
toward the CCN properties of organic particles. Additional studies are needed to confirm LLPS
in SOA particles produced using more atmospherically relevant particle mass concentrations
and submicron sizes. Also, additional investigations are required to confirm LLPS in SOA
particles derived from more complex VOCs.

**Author contribution**

M.S. and H.J.L. conceived and designed the experiments. S.H., J.B.L., and Z.B.B. performed
the experiments and analyzed the data. S.H. and M.S. wrote the manuscript and J.B.L. and
H.J.L. edited the manuscript.





**Acknowledgement**

This work was supported by the National Research Foundation of Korea grant funded by the Korea Government (MSIP) (2016R1C1B1009243). This research was supported by the National Strategic Project-Fine Particle of the National Research Foundation of Korea (NRF) funded by the Ministry of Science and ICT (MSIT), the Ministry of Environment (ME), and the Ministry of Health and Welfare (MOHW) (2017M3D8A1092015).

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





Table 1. Experimental conditions for production and collection of SOA particles from α-pinene ozone (termed 'α-pinene O₃') and photo-oxidation (termed 'α-pinene OH'). The separation relative humidity (SRH) upon moistening and the merging relative humidity (MRH) upon drying are listed. The SRH is the RH at which liquid-liquid phase separation occurred. The MRH is the RH at which two phases merged into one phase. The uncertainties indicates the 2σ from several humidity cycles for one sample and from the uncertainty of the calibration. SRH = 0 and MRH = 0 indicate that phase separation was not observed.

| SOA sample | α-pinene conc. (ppb) | OH exposure (day) | O₃ conc. (ppb) | NH₃ conc. (ppb) | SRH (%) | MRH (%) |
|---|---|---|---|---|---|---|
| α-pinene O₃ #1 | 1000 | - | 10000 | 0 | 96.0 ± 2.3 | 94.3 ± 3.1 |
| α-pinene O₃ #2 | 1000 | - | 10000 | 0 | 95.4 ± 2.0 | 91.6 ± 4.4 |
| α-pinene OH #1 | 1000 | 0.5 | 2000 | 0 | 0 | 0 |
| α-pinene OH #2 | 1000 | 2.5 | 6000 | 0 | 0 | 0 |
| α-pinene OH #3 | 1000 | 0.5 | 2000 | 0 | 0 | 0 |
| α-pinene OH #4 | 1000 | 2.5 | 6000 | 0 | 0 | 0 |





Table 2. Experimental conditions for production and collection of SOA particles from α-pinene ozone with NH₃ (termed 'α-pinene O₃/NH₃') and photo-oxidation with NH₃ (termed 'α-pinene OH/NH₃'). The separation relative humidity (SRH) upon moistening and the merging relative humidity (MRH) upon drying are listed. The SRH is the RH at which liquid-liquid phase separation occurred. The MRH is the RH at which two phases merged into one phase. The uncertainties indicates the 2σ from several humidity cycles for one sample and from the uncertainty of the calibration. SRH = 0 and MRH = 0 indicate that LLPS was not observed.

| SOA sample | α-pinene conc. (ppb) | OH exposure (day) | O₃ conc. (ppb) | NH₃ conc. (ppb) | SRH (%) | MRH (%) |
|---|---|---|---|---|---|---|
| α-pinene O₃/NH₃ #1 | 1000 | - | 10000 | 2000 | 95.4 ± 3.0 | 94.0 ± 2.6 |
| α-pinene O₃/NH₃ #2 | 1000 | - | 10000 | 2000 | 95.4 ± 3.4 | 95.1 ± 2.1 |
| α-pinene OH/NH₃ #1 | 1000 | 2.5 | 6000 | 2000 | 0 | 0 |
| α-pinene OH/NH₃ #2 | 1000 | 0.5 | 2000 | 2000 | 0 | 0 |



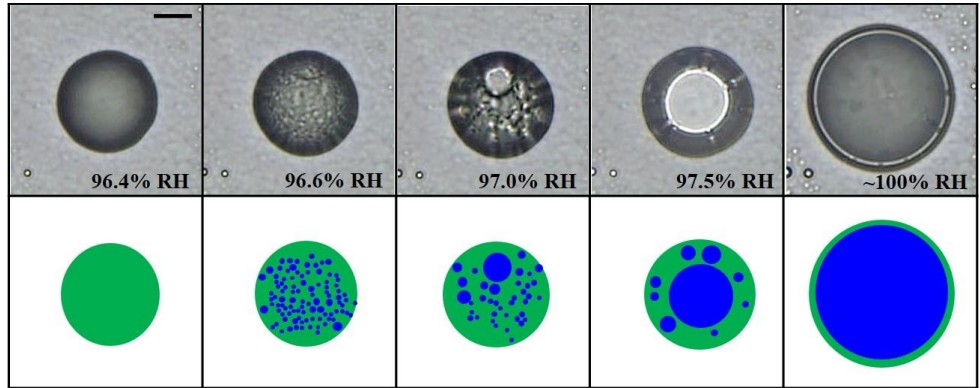

Figure 1. Optical images of a SOA particle produced from α-pinene ozonolysis (α-pinene O$_3$ #1 in Table 1) with increasing RH. Illustrations is for clarifying the image. Green is SOA rich phase, and blue is water rich phase. Scale bar is 20 μm.

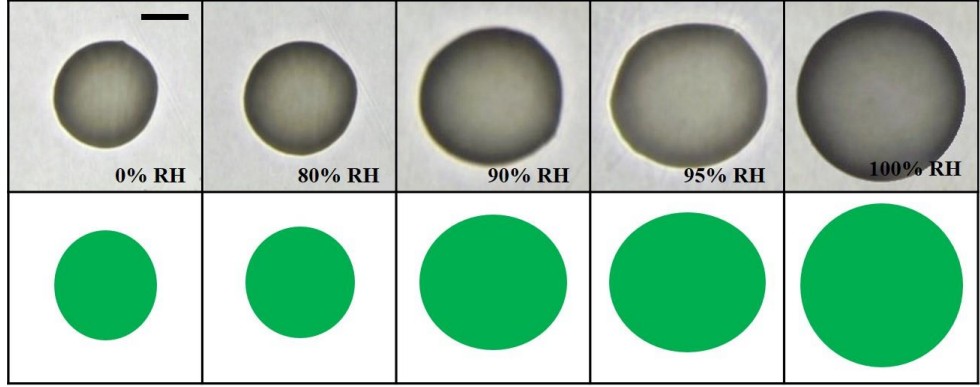

Figure 2. Optical images of a SOA particle produced from α-pinene photo-oxidation (α-pinene OH #2 in Table 1) with increasing RH. Illustrations is for clarifying the image. Green is SOA rich phase, and blue is water rich phase. Scale bar is 20 μm.



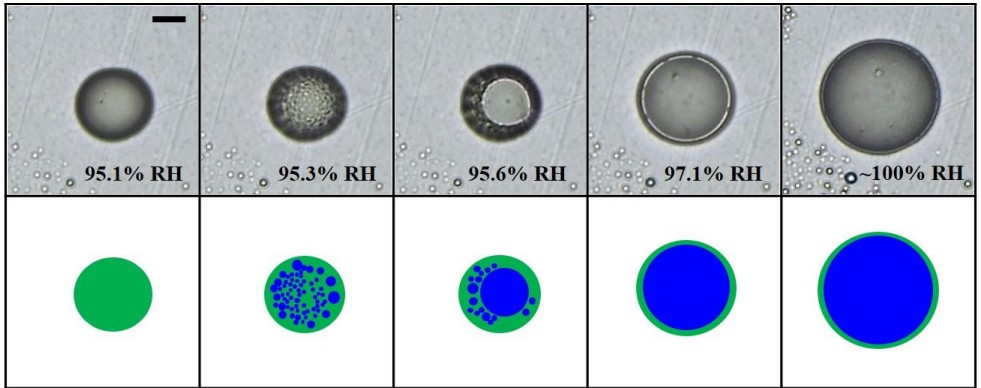

Figure 3. Optical images of a SOA particle produced from α-pinene ozonolysis with NH₃ (α-pinene O₃/NH₃ #1 in Table 2) with increasing RH. Illustrations is for clarifying the image. Green is SOA rich phase, and blue is water rich phase. Scale bar is 20 μm.

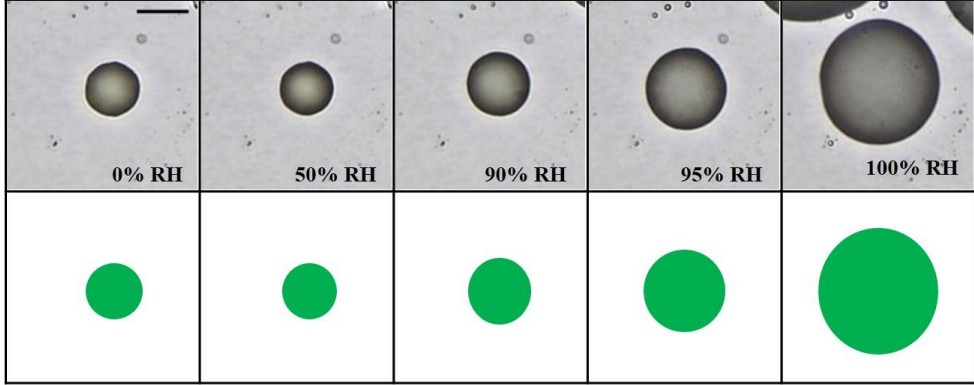

Figure 4. Optical images of a SOA particle produced from α-pinene photo-oxidation with NH₃ (α-pinene OH/NH₃ #2 in Table 1) with increasing RH. Illustrations is for clarifying the image. Green is SOA rich phase, and blue is water rich phase. Scale bar is 20 μm.





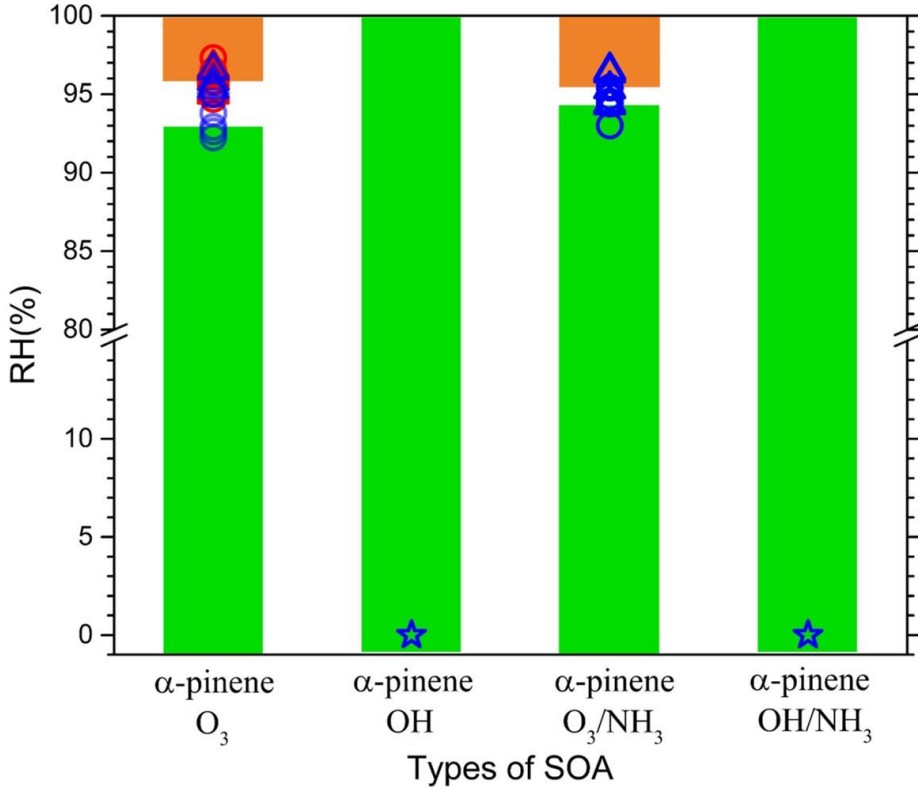

Figure 5. Relative humidity (RH) at which two phases were observed during RH scanning as a function of four different types of SOA particles. Blue and red symbols are from this study and from Renbaum-Wolff et al. (2016), respectively. Circles represent merging RH (MRH) for RH decreasing and triangles represent separation RH (SRH) for RH increasing. RH = 0 % indicates no LLPS. Green shaded region indicates one phase present and orange shaded region indicates two phases present in the SOA particles.




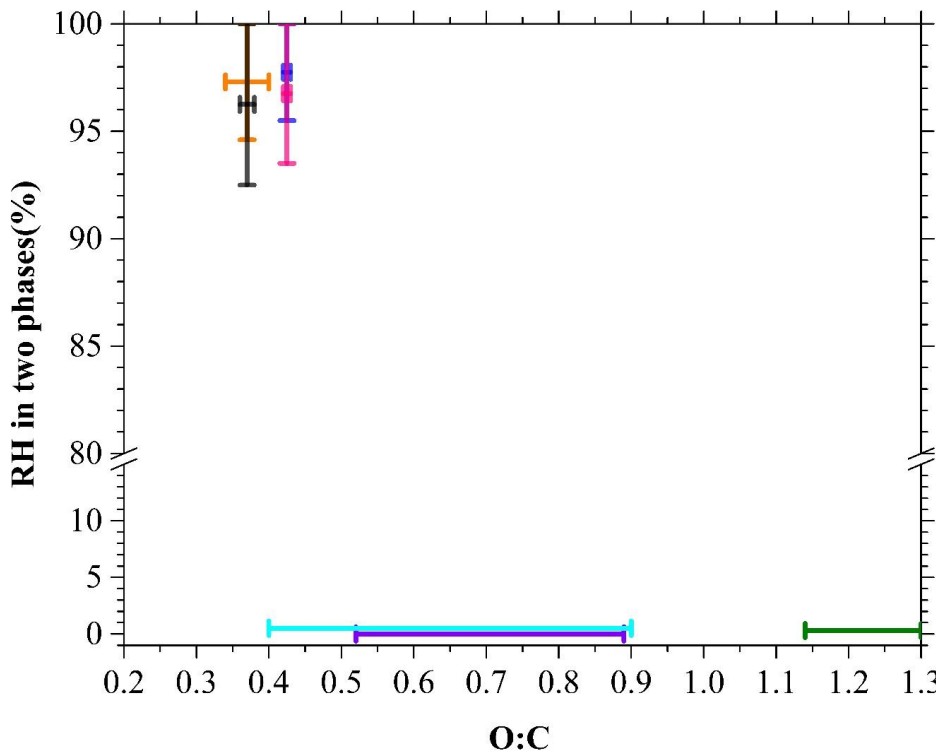

Figure 6. Relative humidity in two phases as a function of average O:C of SOA particles derived from α-pinene ozonolysis (pink) and α-pinene photo-oxidation (cyan) from this study, β-caryophyllene ozonolysis (black) from Song et al. (2017), α-pinene ozonolysis (blue) from Renbaum-Wolff et al. (2016), limonene ozonolysis (orange) from Song et al. (2017), toluene photo-oxidation (green) from Song et al. (2017), and isoprene photo-oxidation (puple) from Rastak et al. (2017). The O:C and related experimental conditions are summarized in Table S1.