# Peer review of "Liquid-liquid phase separation in secondary organic aerosol particles"

_Atmospheric Chemistry and Physics, 2019_

## Referee Comment (RC1) · Anonymous Referee #1 · 28 Feb 2019

The paper "Liquid-liquid phase separation in secondary organic aerosol particles produced from alpha-pinene ozonolysis and alpha-pinene photo-oxidation with/without ammonia" by Ham et al. characterizes the presence or absence of LLPS at high relative humidities under certain experimental conditions, as expressed in the title. The data are compelling and the writing is clear. This study builds directly on the corresponding author's prior work in this area. These systems are expected to impact the formation of CCN, and as a result, are relevant to the audience of Atmospheric Chemistry and Physics. My one main concern about the paper is that the conclusions are

too far reaching. In particular, the authors relate their findings to the O:C ratio needed for LLPS without measuring the O:C ratio for their systems.

major comments Section 3.4: How much does the O:C vary for oxidation in different chambers or flow reactors under the same reactant concentrations? I am skeptical that conclusions about O:C and LLPS can be reached in this paper because there is no direct measure of the O:C of the aerosol particles.

How relevant is it to have aerosol particles just composed of secondary organic material with no other species?

minor comments pg 3, line 2: You could add Rastak et al. 2017, Hodas et al. 2016, Altaf et al. ACS Earth & Space Chem 2018 here as well.

pg 3 line 12: Really, too few systems (5) have been studied so far to define these boundaries in comparison to the number of systems explored for LLPS in the presence of inorganic salts.

pg 4 line 14: I don't understand these two different timescales. Do these correspond to different residence times in the flow cell? Only one number is given for residence time.

pg 4: How similar are the concentrations of alpha-pinene and ammonia to atmospheric concentrations?

pg 4: How much O3 is expected to be converted to OH? What is the concentration of O3 in the flow reactor? What are the expected rate coefficients for alpha-pinene oxidation with OH vs. O3?

pg 6, line 12: I don't understand how the written sentences leads to this subset of references on LLPS. Be more specific about the "consistent results" that this paper has in common with those cited.

pg 6, line 18: Is the mass concentration at SOA provided for a specific RH?

---

## Referee Comment (RC2) · Anonymous Referee #2 · 4 Mar 2019

This manuscript studied the liquid liquid phase separation criteria of alpha-pinene derived SOA from both ozonolysis and photo-oxidation pathways with and without the exposure of ammonia gas. The results show that only the ozonolysis pathway could generate LLPS at high relative humidity, regardless of ammonia exposure or not. The manuscript is an extension of the author's previous work and the results are interesting. However, the lack of direct measurements of the chemical composition of the SOA makes it more difficult and less convincing to justify the conclusions that the authors made. I suggest the authors either include more evidence or modify the conclusions

based on existing evidence before publishing the manuscript. I outline some comments below for the manuscript.

Major comments:

1. The author stated that "The O:C ratio for the SOA particles derived from $\alpha$-pinene ozonolysis ranges from 0.42–0.44 as per Li et al. (2015), whereas that for SOA particles derived 10 from $\alpha$-pinene photo-oxidation is 0.40–0.90 according to Lambe et al. (2015)." And then the author makes the conclusion that "LLPS occurred when the average O:C ratio was between 0.34 and 0.44. However, LLPS did not occur when the average O:C ratio was between 0.40 and 1.30." Please note that the regions of O:C ratios between the LLPS and non-LLPS are overlapping, which makes the conclusion confusing. The author should try to narrow down the O:C ratio for the non-LLPS regions. I recall that the paper by Lambe et al. 2015 shows the O:C ratio based on different OH exposure times. So maybe the authors can compare the OH exposure time in this study with that of the Lambe et al. to obtain a more precise O:C value for photo-oxidation of a-pinene in the the flow tube.

2. Since there is no direct measurement of the chemical composition of the SOA generated from this study, the authors should include more research results to back up the O:C ratios for a-pinene SOA under ozonolysis and photo-oxidation. For instance, Shilling, et al 2009 (http://www.atmos-chem-phys.net/9/771/2009/) shows the O:C values of ozonolysis SOA are 0.3-0.45; Zhang et al 2015 (https://doi.org/10.5194/acp-15-7819-2015, 2015.) used flow tube studies which is similar to the authors' setup, and shows the O:C values are 0.42-0.45; Chen et al. (www.atmos-chem-phys.net/13/5017/2013) used PAM and shows the photo-oxidation SOA has O:C values of 0.6-0.9, all of which are different from the literature values the authors provided in the manuscript. More of these past literatures data need to be included to provide a more convincing conclusion of the O:C values of a-pinene SOA since no actual measurement was made during the experiment.

3. The authors' results seem to imply that whether or not adding ammonia, it would not change the LLPS within the range of the error bars within the reaction timescale of this experiment. However, this result was not included in the conclusion part. The author should state this result more clearly.

4. The author states that O:C values have an influence on the LLPS. How about H:C values? Did any literature suggest that H:C values can alter the LLPS as well?

5. The experimental conditions were not very clear and detailed. Table 1 needs to include more information such as the mode diameter of the particles generated under each situation and the mass concentrations of the particles.

6. The particles generated from the flow tube should be submicron, however the authors described the size collected on the substrate was 1-5 um. Why would be such difference between the particles suspended and on the substrate? I suppose it was due to impaction of the plate. How would this morphology change affect the LLPS process?Have you compared with size values from past AFM and SEM studies performed by Andrew Ault et al.?

Minor comment:

1. The author states that the error bar of the relative humidity control system is 2%, however the results show that the LLPS occurs at 95.8+/- 2.3% and 95.4+/- 2.9%. Since the system has an intrinsic error of 2%, the error bar of the final results should not be down to one decimal point. The results should round up and end at 96% and 95%.

2. The author should specify more in detail how the OH concentrations were calculated. Was it using the rate constant from Eqns 1 and 2? How would the high concentration of a-pinene vapor (1000 ppb) affect the calcluation of the OH concentration?
* * *

---

## Referee Comment (RC3) · Anonymous Referee #3 · 15 Mar 2019

This work investigate the liquid liquid phase separation of alpha-pinene derived SOA from both ozonolysis and photo-oxidation in the presence or absence of ammonia gas. This work shows that only reaction products originated from ozonolysis of alpha-pinene undergo phase separation at a very high humidity. The authors suggest that bulk elemental composition (e.g. O/C ratio) could be a good proxy for determining the phase separation of alpha-pinene derived SOA formed from ozonolysis and photo-oxidation. This work provide valuable data which allow us to better understand the phase state and morphology of ambient SOA. I have a few comments for the authors' consideration.

[Figure]

Comments:

Abstract, "LLPS occurred when the O:C ratio was less than ~0.44 and LLPS did not occur when the O:C ratio was greater than ~0.40. When LLPS was observed". Since the elemental ratios of the SOAs have not been measured in this work, it may not be appropriate to put down the O/C ratios in the abstract.

Experiment section, my major comments are how the high gas-phase concentration of alpha-pinene, ozone, OH, ammonia and aerosol mass loading used in this study affect the molecular composition of the SOAs, which would ultimately govern the phase separation of the aerosols. The authors should give more discussion on these aspects.

Page 5, line 13, "During a humidity cycle, RH was reduced from ~100% to 0%, and then, it was increased to ~100% at a rate of 0.5–1.0% RH·min-1 if LLPS was not observed. If LLPS was observed, the RH was reduced from ~100% to ~5–10% lower than the RH at which the two liquid phases merged into one phase, followed by an increase to ~100% RH at a rate of 0.1–0.5% RH·min-1." Could the authors elaborate how they could confirm an equilibrium state could be achieved for all systems under these RH increasing or decreasing rate?

Page 6, line 17, As shown Table 1, the phase separation was observed at very high RH. Could the authors explain why this happens for the investigated systems?

Also, in addition to the comments on the O/C ratios, could the authors explain why the phase separation does not occur for the SOAs generated from photooxidation of alpha-pinene (e.g. from a molecular insight or perspective)?

Page 9, line 16, "The range of O:C ratio corresponding to the absence of LLPS is wider 17 than that reported by a previous work (Song et al., 2017). The difference could be attributed to the fact that the SOA particles were generated from different types of VOCs." Could the authors elaborate this point a more?

Page 10, line 19, "Moreover, LLPS occurred in the SOA particles at high RH (as high as

∼100%), implying that these results can provide additional information toward the CCN properties of organic particles." Could the authors elaborate what addition information could be gained from the results of this work? How this information would help us to better understand the CCN properties of the particles?

---

## Author Response (AR1)

1  Jason Surratt,

2  Co-Editor of ACP

5  Dear Jason Surratt,

7  Listed below are our responses to the comments from the reviewers of our manuscript. We

8  thank the reviewers for carefully reading our manuscript and for their very helpful suggestions!

9  For clarity and visual distinction, the referee comments or questions are listed here in black

10 and are preceded by bracketed, italicized numbers (e.g. *[1]*). Authors' responses are in red

11 below each referee statement with matching numbers (e.g. *[A1]*).

13 Sincerely,

15 Mijung Song

16 Assistant Professor of Earth and Environmental Sciences

17 Chonbuk National University

21 **Response to Referee #1 (Reviewer comments in black text)**

22

23 The paper "Liquid-liquid phase separation in secondary organic aerosol particles produced

24 from α-pinene ozonolysis and α-pinene photo-oxidation with/without ammonia" by Ham et al.

25 characterizes the presence or absence of LLPS at high relative humidities under certain

26 experimental conditions, as expressed in the title. The data are compelling and the writing is

27 clear. This study builds directly on the corresponding author's prior work in this area. These

28 systems are expected to impact the formation of CCN, and as a result, are relevant to the

29 audience of Atmospheric Chemistry and Physics. My one main concern about the paper is that

30 the conclusions are too far reaching. In particular, the authors relate their findings to the O:C

31 ratio needed for LLPS without measuring the O:C ratio for their systems.

32

33 **Major comments**

34 *[1]* How much does the O:C vary for oxidation in different chambers or flow reactors under

35 the same reactant concentrations? I am skeptical that conclusions about O:C and LLPS can be

36 reached in this paper because there is no direct measure of the O:C of the aerosol particles.

37 How relevant is it to have aerosol particles just composed of secondary organic material with

no other species?

*[A1]* Thank you for the comment. It is difficult to answer since there is no reference showing O:C ranges for SOA generated from α-pinene photo-oxidation under the same reactant concentrations using different chambers or flow reactors. Lambe et al. (2015) compared the O:C ranges of SOA particles produced from α-pinene photo-oxidation using different chambers and flow reactors (Fig. 2a in Lambe et al., 2015) although the experimental conditions were not exactly same. The O:C varies in the range of 0.3 - 1.0 depending on experimental conditions and chambers/reactors. In our study, we chose the O:C ranges (0.4 - 0.9) in the literature that were the closest to our experimental condition.

As shown in Fig. 6 in the manuscript, there appears to be a relationship between the occurrence of LLPS and the O:C of the SOA particles. This behavior was also observed in bulk solutions containing two organics and water (Ganbavale et al., 2015). We will add sentences below in the revised manuscript (pg. 10, lines: 3-8). An additional study on O:C limit for occurrence and absence of LLPS of organic aerosol particles will come up very soon. However, as the three referees pointed out that the O:C conclusion for LLPS is too strong at this stage, we will put down the values in the abstract.

"Similar to the results of LLPS in the SOA particles with O:C ratio, bulk solutions containing two organics and water also showed the miscibility gap (Ganbavale et al., 2015). For example, bulk solutions of two organics with a low O:C and water (e.g. a mixture of 1-butanol, 1-propanol, and water) formed two liquid phases (Ganbavale et al., 2015). However, bulk solutions of two organics with a high O:C and water (e.g. a mixture of ethanol, acetic acid, and water) formed a single liquid phase."

As mentioned in Introduction section (pg. 2, lines: 4-5), it has been found that SOA particles comprise from 20 to 80 % of ultrafine aerosol particles depending on the location (Zhang et al., 2007; Jimenez et al., 2009).

References:

Ganbavale, G., Zuend, A., Marcolli, C., and Peter, T.: Improved AIOMFAC model parameterisation of the temperature dependence of activity coefficients for aqueous organic mixtures, Atmos. Chem. Phys., 15, 447-493, 10.5194/acp-15-447-2015, 2015.

Jimenez, J. L., Canagaratna, M. R., Donahue, N. M., Prevot, A. S. H., Zhang, Q., Kroll, J. H., DeCarlo, P. F., Allan, J. D., Coe, H., Ng, N. L., Aiken, A. C., Docherty, K. S., Ulbrich, I. M., Grieshop, A. P., Robinson, A. L., Duplissy, J., Smith, J. D., Wilson, K. R., Lanz, V. A., Hueglin, C., Sun, Y. L., Tian, J., Laaksonen, A., Raatikainen, T., Rautiainen, J., Vaattovaara, P., Ehn, M., Kulmala, M., Tomlinson, J. M., Collins, D. R., Cubison, M. J., Dunlea, E. J., Huffman, J. A., Onasch, T. B., Alfarra, M. R., Williams, P. I., Bower, K., Kondo, Y., Schneider, J., Drewnick, F., Borrmann, S., Weimer, S., Demerjian, K., Salcedo, D., Cottrell, L., Griffin, R., Takami, A., Miyoshi, T., Hatakeyama, S., Shimono, A., Sun, J. Y., Zhang, Y. M., Dzepina, K., Kimmel, J. R., Sueper, D., Jayne, J. T., Herndon, S. C., Trimborn, A. M., Williams, L. R., Wood, E. C., Middlebrook, A. M., Kolb, C. E., Baltensperger, U., and Worsnop, D. R.: Evolution of Organic Aerosols in the Atmosphere, Science, 326, 1525-1529, DOI 10.1126/science.1180353, 2009.

Lambe, A. T., Chhabra, P. S., Onasch, T. B., Brune, W. H., Hunter, J. F., Kroll, J. H., Cummings, M. J., Brogan, J. F., Parmar, Y., Worsnop, D. R., Kolb, C. E., and Davidovits, P.: Effect of oxidant concentration, exposure time, and seed particles on secondary organic aerosol chemical composition and yield, Atmos. Chem. Phys., 15, 3063-3075, 10.5194/acp-15-3063-2015, 2015.

Zhang, Q., Jimenez, J. L., Canagaratna, M. R., Allan, J. D., Coe, H., Ulbrich, I., Alfarra, M. R., Takami, A., Middlebrook, A. M., Sun, Y. L., Dzepina, K., Dunlea, E., Docherty, K., DeCarlo, P. F., Salcedo, D., Onasch, T., Jayne, J. T., Miyoshi, T., Shimono, A., Hatakeyama, S., Takegawa, N., Kondo, Y., Schneider, J., Drewnick, F., Borrmann, S., Weimer, S., Demerjian, K., Williams, P., Bower, K., Bahreini, R., Cottrell, L., Griffin, R. J., Rautiainen, J., Sun, J. Y., Zhang, Y. M., and Worsnop, D. R.: Ubiquity and dominance of oxygenated species in organic aerosols in anthropogenically-influenced Northern Hemisphere midlatitudes, Geophys. Res. Lett., 34, Artn L13801Doi 10.1029/2007gl029979, 2007.

**Minor comments**

*[2]* pg 3, line 2: You could add Rastak et al. 2017, Hodas et al. 2016, Altaf et al. 2018 here as well.

*[A2]* Thank you. These references will be included in the revised manuscript.

*[3]* pg 3 line 12: Really, too few systems have been studied so far to define these boundaries in comparison to the number of systems explored for LLPS in the presence of inorganic salts.

*[A3]* We agree the referee's comment that only a few systems have been studied on the effect of O:C for occurrence of LLPS in organic aerosols compared to studies for LLPS in the presence of inorganic salts. To address the referee's comment, we will add this point to the revised manuscript in Introduction (pg. 3, lines: 18-19).

"Since still a few systems have been studied so far, more studies are needed to confirm the effect of O:C for LLPS in organic particles."

*[4]* pg 4 line 14: I don't understand these two different timescales. Do these correspond to different residence times in the flow cell? Only one number is given for residence time.

*[A4]* These are OH exposures that correspond to the atmospheric aging time of 0.5 day and 2.5 day as shown in Table 1. Atmospheric aging time was determined using OH radical concentration in the flow reactor, atmospheric OH radical concentration ($1.5 \times 10^6$ molecules $cm^{-3}$), and residence time in the flow reactor (3.63 min). OH concentrations in the flow reactor were calculated using first order photochemical decay of toluene with OH radical (Babar et al., 2017). The concentration of OH radicals was estimated from the photochemical corrosion of toluene because toluene is well known for its OH reaction rate $5.48 \times 10^{-12}$ molecules $cm^{-3}$ $s^{-1}$ with insignificant reaction rate with $O_3$ (Atkinson and Aschmann, 1989). This will be included in Sect. 2.1 in the revised manuscript.

References:

Atkinson, R. and Aschmann, S. M.: Rate constants for the gas-phase reactions of the OH radical with a series of aromatic hydrocarbons at $296 \pm 2$ K, Int. J. Chem. Kinet., 21(5), 355–365, doi:10.1002/kin.550210506, 1989.

Babar, Z. B., Park, J.-H., and Lim, H.-J.: Influence of $NH_3$ on secondary organic aerosols from the ozonolysis and photooxidation of α-pinene in a flow reactor, Atmos. Environ., 164, 71-84, 10.1016/j.atmosenv.2017.05.034, 2017.

*[5]* pg 4: How similar are the concentrations of α-pinene and ammonia to atmospheric concentrations?

*[A5]* α-pinene and ammonia concentrations have been measured in the range of ~10 - 600 ppt (Kim et al., 2005; Jaars et al., 2018) and 0.3 - 120 ppb (Carmichael et al., 2003; Meng et al., 2011; Artíñano et al., 2018; Song et al., 2018; Kumar et al., 2019), respectively, in different environments (i.e. forest and polluted regions). Compared to the concentrations of α-pinene and ammonia in the atmosphere, much higher concentrations of the α-pinene and ammonia were used in this study due to the experimental constraints for SOA generation. Further studies are needed to confirm LLPS in SOA particles produced more atmospherically relevant the VOC mass concentrations. We will address this point in Summary Sect. (pg. 11, lines: 18-21).

References:

Artíñano, B., Pujadas, M., Alonso-Blanco, E., Becerril-Valle, M., Coz, E., Gómez-Moreno, F. J., Salvador, P., Nuñez, L., Palacios, M. and Diaz, E.: Real-time monitoring of atmospheric ammonia during a pollution episode in Madrid (Spain), Atmos. Environ., 189, 80–88, doi:10.1016/j.atmosenv.2018.06.037, 2018.

Carmichael, G. R., Cotrina, J. S., Lacaux, J.-P., Kimani, W., Pienaar, J., Chan, L. ., Thongboonchoo, N., Boonjawat, J., Viet, P. H., Shrestha, A. B., Chen, T., Brunke, E. B., Tavares, T., Bilici, E., Athayde, A., Peng, L. C., Woo, J.-H., Murano, K., Jie, T., Barturen, O., Kirouane, A., Dhiharto, S., Ferm, M., Jose, A. M., Guoan, D., Cerda, J. C., Mohan, M., Bala, R., Mossberg, C., Upatum, P., Harjanto, H., Richard, S. and Adhikary, S. P.: Measurements of sulfur dioxide, ozone and ammonia concentrations in Asia, Africa, and South America using passive samplers, Atmos. Environ., 37(9–10), 1293–1308, doi:10.1016/s1352-2310(02)01009-9, 2003.

Jaars, K., Vestenius, M., Zyl, P. G. Van, Beukes, J. P., Hellén, H., Vakkari, V., Venter, M., Josipovic, M. and Hakola, H.: Receptor modelling and risk assessment of volatile organic compounds measured at a regional background site in South Africa, Atmos. Environ., 172(November 2017), 133–148, doi:10.1016/j.atmosenv.2017.10.047, 2018.

Kim, K., Kim, J., and Lim, J.: Comparison of anthropogenic and natural VOC concentrations in the forest ambient air, J. Kor. Soc. Environ. Anal., 8(3), 132-136, 2005.

Kumar, A., Patil, R. S., Dikshit, A. K. and Kumar, R.: Assessment of Spatial Ambient Concentration of $NH_3$ and its Health Impact for Mumbai City TT, Asian J. Atmos. Environ., 13(1), 11–19 [online] Available from: http://www.dbpia.co.kr/Article/NODE07993471, 2019.

Meng, Z. Y., Lin, W. L., Jiang, X. M., Yan, P., Wang, Y., Zhang, Y. M., Jia, X. F. and Yu, X. L.: Characteristics of atmospheric ammonia over Beijing, China, Atmospheric Chemistry and Physics, Atmos. Chem. Phys., 11(12), 6139–6151, doi:10.5194/acp-11-6139-2011, 2011.

Song, L., Liu, X., Skiba, U., Zhu, B., Zhang, X., Liu, M., Twigg, M., Shen, J., Dore, A., Reis, S., Coyle, M., Zhang, W., Levy, P. and Fowler, D.: Ambient concentrations and deposition rates of selected reactive nitrogen species and their contribution to PM2.5 aerosols at three locations with contrasting land use in southwest China, Environ. Pollut., 233(2), 1164–1176, doi:10.1016/j.envpol.2017.10.002, 2018.

*[6]* pg 4: How much $O_3$ is expected to be converted to OH? What is the concentration of $O_3$ in the flow reactor? What are the expected rate coefficients for α-pinene oxidation with OH vs. $O_3$?

*[A6]* In case of α-pinene photo-oxidation at 10% RH corresponding to the atmospheric aging time of 0.5 d and 2.5 d, residual $O_3$ concentrations in the flow reactor were approximately 1300 ppb and 3600 ppb, respectively, as already mentioned in the main text of the manuscript (pg. 4, lines 18-22). In case of α-pinene photo-oxidation at the atmospheric aging time of 0.5 d and 2.5 d, the OH-reaction rates were 6 and 12 times higher than $O_3$-reaction rates, respectively. The detailed method of SOA particle generation was described previously by Babar et al. (2017).

Reference:

Babar, Z. B., Park, J.-H., and Lim, H.-J.: Influence of $NH_3$ on secondary organic aerosols from the ozonolysis and photooxidation of α-pinene in a flow reactor, Atmos. Environ., 164, 71-84, 10.1016/j.atmosenv.2017.05.034, 2017.

*[7]* pg 6, line 12: I don't understand how the written sentences leads to this subset of references

on LLPS. Be more specific about the "consistent results" that this paper has in common with those cited.

*[A7]* The references will be relocated in the revised manuscript to make this point clear (pg. 6, lines: 21-26).

"Moreover, previous studies using surface tension (Jasper, 1972), spreading coefficient (Kwamena et al., 2010; Reid et al., 2011), Raman spectroscopy (Song et al., 2013; Gorkowski et al., 2016; Gorkowski et al., 2017), atomic force microscopy (Zhang et al., 2018), and scanning electron microscopy (O'Brien et al., 2015) showed consistent results with regard to the morphology of the particles consisting of organic and inorganic salts"

Reference:

Gorkowski, K., Beydoun, H., Aboff, M., Walker, J. S., Reid, J. P., and Sullivan, R. C.: Advanced aerosol optical tweezers chamber design to facilitate phase-separation and equilibration timescale experiments on complex droplets, Aerosol. Sci. Tech., 50, 1327-1341, 10.1080/02786826.2016.1224317, 2016.

Gorkowski, K., Donahue, N. M., and Sullivan, R. C.: Emulsified and Liquid Liquid Phase-Separated States of alpha-Pinene Secondary Organic Aerosol Determined Using Aerosol Optical Tweezers, Environ. Sci. Technol., 51, 12154-12163, 10.1021/acs.est.7b03250, 2017.

Jasper, J. J.: The surface tension of pure liquid compounds, J. Phys. Chem. Ref. Data, 1, 841–1009, https://doi.org/10.1063/1.3253106, 1972.

Kwamena, N. O. A., Buajarern, J., and Reid, J. P.: Equilibrium Morphology of Mixed Organic/Inorganic/Aqueous Aerosol Droplets: Investigating the Effect of Relative Humidity and Surfactants, J. Phys. Chem. A., 114, 5787-5795, 10.1021/jp1003648, 2010.

O'Brien, R. E., Wang, B. B., Kelly, S. T., Lundt, N., You, Y., Bertram, A. K., Leone, S. R., Laskin, A., and Gilles, M. K.: Liquid-Liquid Phase Separation in Aerosol Particles: Imaging at the Nanometer Scale, Environ. Sci. Technol., 49, 4995-5002, 10.1021/acs.est.5b00062, 2015.

Reid, J. P., Dennis-Smither, B. J., Kwamena, N. O. A., Miles, R. E. H., Hanford, K. L., and Homer, C. J.: The morphology of aerosol particles consisting of hydrophobic and hydrophilic phases: hydrocarbons, alcohols and fatty acids as the hydrophobic component, Phys. Chem. Chem. Phys., 13, 15559-15572, 10.1039/c1cp21510h, 2011.

Song, M., Marcolli, C., Krieger, U. K., Lienhard, D. M., and Peter, T.: Morphologies of mixed organic/inorganic/aqueous aerosol droplets, Faraday. Discuss., 165, 289-316, 10.1039/c3fd00049d, 2013.

Zhang, Y., Chen, Y. Z., Lambe, A. T., Olson, N. E., Lei, Z. Y., Craig, R. L., Zhang, Z. F., Gold, A., Onasch, T. B., Jayne, J. T., Worsnop, D. R., Gaston, C. J., Thornton, J. A., Vizuete, W., Ault, A. P., and Surratt, J. D.: Effect of the Aerosol-Phase State on Secondary Organic Aerosol Formation from the Reactive Uptake of Isoprene-Derived Epoxydiols (IEPDX), Environ. Sci. Tech. Let., 5, 167-174, 10.1021/acs.estlett.8b00044, 2018.

*[8]* pg 6, line 18: Is the mass concentration at SOA provided for a specific RH?

*[A8]* These are dry SOA mass concentrations at 60% RH measured by SMPS. We will give more information in Experimental Sect. (pg. 6, line: 28 to pg. 7, line: 4).

"A diffusion dryer loaded with silica gel was used at the upstream of Scanning Mobility Particle Sizer (SMPS+C, Grimm, Germany) for the measurement of dry SOA mass concentrations. After dilution, the mass concentrations of the SOA particles were measured to range between ~480 $\mu g \cdot m^{-3}$ and ~880 $\mu g \cdot m^{-3}$ using the SMPS for different experimental conditions as presented in Tables 1 and 2."

**Response to Referee #2 (Reviewer comments in black text)**

This manuscript studied the liquid-liquid phase separation criteria of α-pinene derived SOA from both ozonolysis and photo-oxidation pathways with and without the exposure of ammonia gas. The results show that only the ozonolysis pathway could generate LLPS at high relative humidity, regardless of ammonia exposure or not. The manuscript is an extension of the author's previous work and the results are interesting. However, the lack of direct measurements of the chemical composition of the SOA makes it more difficult and less convincing to justify the conclusions that the authors made. I suggest the authors either include more evidence or modify the conclusions based on existing evidence before publishing the manuscript. I outline some comments below for the manuscript.

**Major comments**

*[1]* The author stated that "The O:C ratio for the SOA particles derived from pinene ozonolysis ranges from 0.42–0.44 as per Li et al. (2015), whereas that for SOA particles derived from α-pinene photo-oxidation is 0.40–0.90 according to Lambe et al. (2015)." And then the author makes the conclusion that "LLPS occurred when the average O:C ratio was between 0.34 and 0.44. However, LLPS did not occur when the average O:C ratio was between 0.40 and 1.30." Please note that the regions of O:C ratios between the LLPS and non-LLPS are overlapping, which makes the conclusion confusing. The author should try to narrow down the O:C ratio for the non-LLPS regions. I recall that the paper by Lambe et al. 2015 shows the O:C ratio based on different OH exposure times. So maybe the authors can compare the OH exposure time in this study with that of the Lambe et al. to obtain a more precise O:C value for photo-oxidation of a-pinene in the flow tube.

*[A1]* Thank you for the comment. The referee suggested to narrow down the O:C regions for the non-LLPS due to O:C overlapping. However, we would like to keep it with the ranges at this stage because 1) the O:C ranges (0.4 - 0.9) were from literature (Lambe et al., 2015, exposure time: 0.2 – 17 days) that was the closest to our experimental condition (not our direct measurement), and 2) previous studies have also showed O:C region overlapping for LLPS and non-LLPS (Renbaum-Wolff et al., 2016; Rastak et al., 2017; Song et al., 2017). An additional study on O:C limit for occurrence and absence of LLPS of organic aerosol particles will come up very soon.

References:

Lambe, A. T., Chhabra, P. S., Onasch, T. B., Brune, W. H., Hunter, J. F., Kroll, J. H., Cummings, M. J., Brogan, J. F., Parmar, Y., Worsnop, D. R., Kolb, C. E., and Davidovits, P.: Effect of oxidant concentration, exposure time, and seed particles on secondary organic aerosol chemical composition and yield, Atmos. Chem. Phys., 15, 3063-3075, 10.5194/acp-15-3063-2015, 2015.

Rastak, N., Pajunoja, A., Navarro, J. C. A., Ma, J., Song, M., Partridge, D. G., Kirkevag, A., Leong, Y., Hu, W. W., Taylor, N. F., Lambe, A., Cerully, K., Bougiatioti, A., Liu, P., Krejci, R., Petaja, T., Percival, C., Davidovits, P., Worsnop, D. R., Ekman, A. M. L., Nenes, A., Martin, S., Jimenez, J. L., Collins, D. R., Topping, D. O., Bertram, A. K., Zuend, A., Virtanen, A., and Riipinen, I.: Microphysical explanation of the RH-dependent water affinity of biogenic organic aerosol and its importance for climate, Geophys. Res. Lett., 44, 5167-5177, 2017.

Renbaum-wolff, L., Song, M., Marcolli, C., Zhang, Y., and Liu, P. F.: Observations and implications of liquid − liquid phase separation at high relative humidities in secondary organic material produced by α -pinene ozonolysis without inorganic salts, Atmos. Chem. Phys., 16, 7969-7979, 10.5194/acp-16-7969-2016, 2016.

Song, M., Liu, P. F., Martin, S. T., and Bertram, A. K.: Liquid-liquid phase separation in particles containing secondary organic material free of inorganic salts, Atmos. Chem. Phys., 17, 11261-11271, 2017.

*[2]* Since there is no direct measurement of the chemical composition of the SOA generated from this study, the authors should include more research results to back up the O:C ratios for a-pinene SOA under ozonolysis and photo-oxidation. For instance, Shilling, et al 2009 (http://www.atmos-chem-phys.net/9/771/2009/) shows the O:C values of ozonolysis SOA are 0.3-0.45; Zhang et al 2015 (https://doi.org/10.5194/acp- 15-7819-2015, 2015.) used flow tube studies which is similar to the authors' setup, and shows the O:C values are 0.42-0.45; Chen et al. (www.atmos-chemphys. net/13/5017/2013) used PAM and shows the photo-oxidation SOA has O:C values of 0.6-0.9, all of which are different from the literature values the authors provided in the manuscript. More of these past literatures data need to be included to provide

a more convincing conclusion of the O:C values of a-pinene SOA since no actual measurement was made during the experiment.

*[A2]* Thank you for the comment. We fully agree the Reviewer's comment. As the three Reviewers suggested, we will add more references for O:C ratio of SOA particles produced by α-pinene ozonolysis and α-pinene photo-oxidation using flow reactors or flow tube reactors under similar reactant concentrations in the revised manuscript (Table S1).

*[3]* The authors' results seem to imply that whether or not adding ammonia, it would not change the LLPS within the range of the error bars within the reaction timescale of this experiment. However, this result was not included in the conclusion part. The author should state this result more clearly.

*[A3]* Thank you for the suggestion! To make it clearer, we will include sentences in Summary Sect (pg. 11, lines: 9-11).

"LLPS occurred in the SOA particles produced by α-pinene ozonolysis while no LLPS was observed in the SOA particles produced by α-pinene photo-oxidation. In addition, the occurrence of LLPS did not depend on the presence and absence of $NH_3$."

*[4]* The author states that O:C values have an influence on the LLPS. How about H:C values? Did any literature suggest that H:C values can alter the LLPS as well?

*[A4]* In previous studies of Song et al. (2012) and You et al. (2013), they showed no relationship with H:C for occurrence of LLPS in particles containing organic and inorganic salts.

References:
Song, M., Marcolli, C., Krieger, U. K., Zuend, A., and Peter, T.: Liquid-liquid phase separation in aerosol particles: Dependence on O:C, organic functionalities, and compositional complexity, Geophys. Res. Lett., 39, 1-5, 10.1029/2012GL052807, 2012.
You, Y., Renbaum-Wolff, L., and Bertram, A. K.: Liquid-liquid phase separation in particles containing organics mixed with ammonium sulfate, ammonium bisulfate, ammonium

nitrate or sodium chloride, Atmos. Chem. Phys., 13, 11723-11734, 10.5194/acp-13-11723-2013, 2013.

*[5]* The experimental conditions were not very clear and detailed. Table 1 needs to include more information such as the mode diameter of the particles generated under each situation and the mass concentrations of the particles.

*[A5]* As suggested, Tables 1 and 2 will be updated in the revised manuscript to include geometric mean diameter and mass concentration of particles for each experimental condition.

*[6]* The particles generated from the flow tube should be submicron, however the authors described the size collected on the substrate was 1-5 um. Why would be such difference between the particles suspended and on the substrate? I suppose it was due to impaction of the plate. How would this morphology change affect the LLPS process? Have you compared with size values from past AFM and SEM studies performed by Andrew Ault et al.?

*[A6]* In this study, the SOA particles during generation were collected on a hydrophobic substrate at the outlet of the flow tube reactor. During the particle collection time, the SOA particles might coagulate on the substrate resulting in larger particles consisting of up to ~5 μm. To make it clearer, an optical image of the SOA particles on a hydrophobic substrate at the outlet of the flow tube reactor will be included in Supplement (Fig. S1).

For the LLPS experiments, supermicron particles consisting of $20 - 80$ μm in diameter are required since the microscope is equipped with a long working distance objective and a flow-cell. In order to obtain the appropriate particle sizes for the LLPS experiments, the SOA particles collected on the substrate at the outlet of the flow tube reactor underwent a process of growth and coagulation at ~100 % RH. The detailed method of producing supermicron particles was described previously in Renbaum-Wolff et al. (2016) and Song et al. (2015). In our study, we did not observe a dependence of LLPS on the particle size across the studied range. Ault et al. (2013) also showed that LLPS occurred in sea spray aerosol particles consisting of $0.3 - 2$ μm. We will include this reference in Introduction Sect (pg. 2, line 23).

References:

Ault, A. P., Guasco, T. L., Ryder, O. S., Baltrusaitis, J., Cuadra-Rodriguez, L. A., Collins, D. B., Ruppel, M. J., Prather, K. A., Bertram, T. H., Grassia, V. H.: Inside versus Outside: Ion Redistribution in HNO3 Reacted Sea Spray Aerosol Particles as Determined by Single Particle Analysis, J. Am. Chem. Soc., 135(39), 14528-14531, 2013.

Renbaum-Wolff, L., Song, M., Marcolli, C., Zhang, Y., and Liu, P. F.: Observations and implications of liquid – liquid phase separation at high relative humidities in secondary organic material produced by α -pinene ozonolysis without inorganic salts, Atmos. Chem. Phys., 16, 7969-7979, 10.5194/acp-16-7969-2016, 2016.

Song, M., Liu, P. F., Hanna, S. J., Li, Y. J., Martin, S. T., and Bertram, A. K.: Relative humidity-dependent viscosities of isoprene-derived secondary organic material and atmospheric implications for isoprene-dominant forests, Atmos. Chem. Phys., 15, 5145-5159, 10.5194/acp-15-5145-2015, 2015.

**Minor comments**

*[7]* The author states that the error bar of the relative humidity control system is 2%, however the results show that the LLPS occurs at 95.8+/- 2.3% and 95.4+/- 2.9%. Since the system has an intrinsic error of 2%, the error bar of the final results should not be down to one decimal point. The results should round up and end at 96% and 95%.

*[A7]* As stated in Sect. 2.2 (pg. 6, line: 5), the uncertainty of the RH after calibration was ±2.0%. The uncertainties of the separation relative humidity (SRH) upon moistening and the merging relative humidity (MRH) upon drying indicate the 2σ from several humidity cycles for one sample and from the uncertainty of the calibration as stated in Table 1 and 2. To make this point clearer, we will add sentences in Sect. 3.1 in the revised manuscript (pg. 6, line: 30 to pg. 7, line: 2).

*[8]* The author should specify more in detail how the OH concentrations were calculated. Was it using the rate constant from Eqns 1 and 2? How would the high concentration of a-pinene vapor (1000 ppb) affect the calculation of the OH concentration?

*[A8]* As suggested, we will add the information of the OH concentrations and the method in the revised manuscript (pg: 4, lines: 13-18):

"In the flow reactor, OH concentrations were determined from the photochemical decay of toluene because toluene is well known for its OH reaction rate. The OH reaction rate constant ($k_{OH}$) of toluene is $5.48 \times 10^{-12}$ molecules cm$^{-3}$ s$^{-1}$ with insignificant reaction rate with $O_3$ (Atkinson and Aschmann, 1989). OH concentrations were calculated by varying $O_3$ and RH from 2000 ppb to 8000 ppb and 10% to 60%, respectively. OH concentrations were calculated by first order decay of toluene by reaction with OH radicals (Babar et al., 2017)"

References:

Atkinson, R. and Aschmann, S. M.: Rate constants for the gas-phase reactions of the OH radical with a series of aromatic hydrocarbons at $296 \pm 2$ K, Int. J. Chem. Kinet., 21(5), 355–365, doi:10.1002/kin.550210506, 1989.

Babar, Z. B., Park, J.-H., and Lim, H.-J.: Influence of $NH_3$ on secondary organic aerosols from the ozonolysis and photooxidation of α-pinene in a flow reactor, Atmos. Environ., 164, 71-84, 10.1016/j.atmosenv.2017.05.034, 2017.

**Response to Referee #3 (Reviewer comments in black text)**

This work investigate the liquid-liquid phase separation of α-pinene derived SOA from both ozonolysis and photo-oxidation in the presence or absence of ammonia gas. This work shows that only reaction products originated from ozonolysis of α-pinene undergo phase separation at a very high humidity. The authors suggest that bulk elemental composition (e.g. O/C ratio) could be a good proxy for determining the phase separation of α-pinene derived SOA formed from ozonolysis and photo-oxidation. This work provide valuable data which allow us to better understand the phase state and morphology of ambient SOA. I have a few comments for the authors' consideration.

**Major comments**

*[1]* Abstract, "LLPS occurred when the O:C ratio was less than ～0.44 and LLPS did not occur when the O:C ratio was greater than ～0.40. When LLPS was observed". Since the elemental ratios of the SOAs have not been measured in this work, it may not be appropriate to put down the O/C ratios in the abstract.

*[A1]* We agree the referee's suggestion. We will remove the value from the Abstract as suggested.

*[2]* Experiment section, my major comments are how the high gas-phase concentration of α-pinene, ozone, OH, ammonia and aerosol mass loading used in this study affect the molecular composition of the SOAs, which would ultimately govern the phase separation of the aerosols. The authors should give more discussion on these aspects.

*[A2]* For the case of α-pinene ozonolysis and photo-oxidation, high gas phase concentration of α-pinene elevates SOA mass loadings and yield by the condensation of low volatile species into particle phase (Cocker et al., 2001; Na et al., 2007; Saathoff et al., 2008; Wang et al., 2014). It has been reported that this did not significantly affect the molecular compositions (Shilling et al., 2008; Shilling et al., 2009; Bertram et al., 2011; Chen et al., 2011; Zuend and Seinfeld, 2012). For instance, Shilling et al. (2009) and Zuend and Seinfeld (2012) showed that the O:C in α-pinene SOA did not depend significantly on mass loadings. Table S1 will be updated with the references in the revised manuscript.

References:

Bertram, A. K., Martin, S. T., Hanna, S. J., Smith, M. L., Bodsworth, A., Chen, Q., Kuwata, M., Liu, A., You, Y., and Zorn, S. R.: Predicting the relative humidities of liquid-liquid phase separation, efflorescence, and deliquescence of mixed particles of ammonium sulfate, organic material, and water using the organic-to-sulfate mass ratio of the particle and the oxygen-to-carbon elemental ratio of the organic component, Atmos. Chem. Phys., 11, 10995-11006, 10.5194/acp-11-10995-2011, 2011.

Chen, Q., Liu, Y., Donahue, N. M., Shilling, J. E., and Martin, S. T.: Particle-Phase Chemistry of Secondary Organic Material: Modeled Compared to Measured O:C and H:C Elemental Ratios Provide Constraints, Environ. Sci. Technol., 45, 4763–4770, doi:10.1021/es104398s, 2011.

Cocker, D. R., Flagan, R. C. and Seinfeld, J. H.: State-of-the-Art Chamber Facility for Studying Atmospheric Aerosol Chemistry, Environ. Sci. Technol., 35(12), 2594–2601, doi:10.1021/es0019169, 2001.

Na, K., Song, C., Switzer, C., and Cocker, D. R.: Effect of ammonia on secondary organic aerosol formation from alpha-Pinene ozonolysis in dry and humid conditions, Environ. Sci. Technol., 41, 6096-6102, 10.1021/es061956y, 2007.

Saathoff, H., Naumann, K.-H., M€ohler, O., Jonsson, Å.M., Hallquist, M., Kiendler-Scharr, A., Mentel, T.F., Tillmann, R., Schurath, U.: Temperature dependence of yields of secondary organic aerosols from the ozonolysis of a-pinene and limonene. Atmos. Chem. Phys. Discuss. 8, 15595e15664. http://dx.doi.org/10.5194/acpd-8-15595-2008, 2008.

Shilling, J. E., Chen, Q., King, S. M., Rosenoern, T., Kroll, J. H., Worsnop, D. R., McKinney, K. A., and Martin, S. T.: Particle mass yield in secondary organic aerosol formed by the dark ozonolysis of α-pinene, Atmos. Chem. Phys., 8, 2073–2088, doi:10.5194/acp-8-2073-2008, 2008.

Shilling, J. E., Chen, Q., King, S. M., Rosenoern, T., Kroll, J. H., Worsnop, D. R., DeCarlo, P. F., Aiken, A. C., Sueper, D., Jimenez, J. L., Martin, S. T.: Loading-dependent elemental composition of α-pinene SOA particles, Atmos. Chem. Phys., 9, 771-782, 10.5194/acp-9-771-2009, 2009.

Wang, X., Liu, T., Bernard, F., Ding, X., Wen, S., Zhang, Y., Zhang, Z., He, Q., Lü, S., Chen, J., Saunders, S., Yu, J.: Design and characterization of a smog chamber for studying gas-phase chemical mechanisms and aerosol formation. Atmos. Meas. Tech. 7, 301e313. http://dx.doi.org/10.5194/amt-7-301-2014, 2014.

Zuend, A., and Seinfeld, J. H.: Modeling the gas-particle partitioning of secondary organic aerosol: the importance of liquid-liquid phase separation, Atmos. Chem. Phys., 12, 3857-3882, 10.5194/acp-12-3857-2012, 2012.

*[3]* Page 5, line 13, "During a humidity cycle, RH was reduced from ～100% to 0%, and then, it was increased to ～100% at a rate of 0.5–1.0% RH·min$^{-1}$ if LLPS was not observed. If LLPS was observed, the RH was reduced from ～100% to ～5–10% lower than the RH at which the two liquid phases merged into one phase, followed by an increase to ～100% RH at a rate of 0.1–0.5% RH·min$^{-1}$." Could the authors elaborate how they could confirm an equilibrium state could be achieved for all systems under these RH increasing or decreasing rate?

*[A3]* This is a good point. At the beginning of LLPS experiments, the SOA particles were equilibrated at ~100% RH for 60 min. Then, a humidity cycle was performed at a rate of 0.1 − 0.5 % RH·min$^{-1}$ for occurrence of LLPS and at a rate of 0.5 − 1.0 % RH·min$^{-1}$ for absence of LLPS. We did not observe a dependence of LLPS and non-LLPS on the RH ramp rate. This point will be included in the revised manuscript (pg. 5, lines: 22-26).

"During a humidity cycle, RH was reduced from ~100% to ~5 − 10% lower than the RH at which the two liquid phases merged into one phase, followed by an increase to ~100% RH at a rate of 0.1 − 0.5% RH·min$^{-1}$. If LLPS was not observed, RH was reduced from ~100% to ~0%, and then, it was increased to ~100% at a rate of 0.5 − 1.0% RH·min$^{-1}$. We did not observe a dependence of LLPS on the humidity ramp rate."

*[4]* Page 6, line 17, As shown Table 1, the phase separation was observed at very high RH. Could the authors explain why this happens for the investigated systems?
Also, in addition to the comments on the O/C ratios, could the authors explain why the phase

separation does not occur for the SOAs generated from photooxidation of α-pinene (e.g. from a molecular insight or perspective)?

*[A4]* As RH increases, water molecules would increase in the particle leaving organic molecules which are less hydrophilic resulting in phase separation in the SOA particles. Previous studies showed that a number of water-soluble organic compounds such as carboxylic acids were identified in the SOA particles produced from α-pinene photo-oxidation (Reinning et al., 2008; Eddingsaas et al., 2012).

References:

Eddingsaas, N. C., Loza, C. L., Yee, L. D., Chan, M., Schilling, K. A., Chhabra, P. S., Seinfeld, J. H., and Wennberg, P. O.: alpha-pinene photooxidation under controlled chemical conditions - Part 2: SOA yield and composition in low- and high-NOx environments, Atmos. Chem. Phys., 12, 7413-7427, 10.5194/acp-12-7413-2012, 2012.

Reinnig, M.C., Müller, L., Warnke, J., Hoffmann, T.: Characterization of selected organic compound classes in secondary organic aerosol from biogenic VOCs by HPLC/MSn, Anal. Bioanal. Chem., 391, 171–182, https://doi.org/10.1007/s00216-008-1964-5, 2008.

*[5]* Page 9, line 16, "The range of O:C ratio corresponding to the absence of LLPS is wider than that reported by a previous work (Song et al., 2017). The difference could be attributed to the fact that the SOA particles were generated from different types of VOCs." Could the authors elaborate this point a more?

*[A5]* Thank you for the comment. To make this point clearer, we will modify the sentences in Sect. 3.4 (pg. 9, line: 29 to pg. 10, line: 2).

"However, LLPS did not occur when the average O:C ratio was between 0.45 and 1.30 in this study. Using a new type of SOA particle generated from α-pinene photo-oxidation, we suggest that the absence of LLPS is wider than that reported by a previous work (0.52 - 1.30) (Song et al., 2017).

Reference:

Song, M., Liu, P. F., Martin, S. T., and Bertram, A. K.: Liquid-liquid phase separation in particles containing secondary organic material free of inorganic salts, Atmos. Chem. Phys., 17, 11261-11271, 2017.

*[6]* Page 10, line 19, "Moreover, LLPS occurred in the SOA particles at high RH (as high as ～100%), implying that these results can provide additional information toward the CCN properties of organic particles." Could the authors elaborate what addition information could be gained from the results of this work? How this information would help us to better understand the CCN properties of the particles?

*[A6]* We fully agree the referee's suggestion. We will address LLPS occurrence and related CCN properties in the revised manuscript (pg. 10 lines: 18-25).

"The two phases consisting of an organic-rich shell and water-rich core were observed at RH as high as ~100% in all cases. Recent studies of Rastak et al. (2017) and Liu et al. (2018) showed from laboratory study and modeling results that the presence of LLPS in organic particles at ~100% RH can lead to lower surface tension, and finally a lower kinetic barrier to CCN activation. Our result can also give an additional insight into attempting more accurate predictions of the CCN properties of organic particles (Petters et al. 2006; Hodas et al. 2016; Renbaum-Wolff et al., 2016; Ovadnevaite et al., 2017; Rastak et al., 2017; Liu et al., 2018)."

References:

Hodas, N., Zuend, A., Schilling, K., Berkemeier, T., Shiraiwa, M., Flagan, R. C., and Seinfeld, J. H.: Discontinuities in hygroscopic growth below and above water saturation for laboratory surrogates of oligomers in organic atmospheric aerosols, Atmos. Chem. Phys., 16, 12767-12792, 10.5194/acp-16-12767-2016, 2016.

Liu, P. F., Song, M., Zhao, T. N., Gunthe, S. S., Ham, S. H., He, Y. P., Qin, Y. M., Gong, Z. H., Amorim, J. C., Bertram, A. K., and Martin, S. T.: Resolving the mechanisms of hygroscopic growth and cloud condensation nuclei activity for organic particulate matter, Nat. Commun., 9, ARTN 4076 10.1038/s41467-018-06622-2, 2018.

Ovadnevaite, J., Zuend, A., Laaksonen, A., Sanchez, K. J., Roberts, G., Ceburnis, D., Decesari, S., Rinaldi, M., Hodas, N., Facchini, M. C., Seinfeld, J. H., and Dowd, C. O.: Surface tension prevails over solute effect in organic-influenced cloud droplet activation, Nature, 546, 637-641, 10.1038/nature22806, 2017.

Petters, M. D., Kreidenweis, S. M., Snider, J. R., Koehler, K. A., Wang, Q., Prenni, A. J., and Demott, P. J.: Cloud droplet activation of polymerized organic aerosol, Tellus. B., 58, 196-205, 10.1111/j.1600-0889, 2006.

Rastak, N., Pajunoja, A., Navarro, J. C. A., Ma, J., Song, M., Partridge, D. G., Kirkevag, A., Leong, Y., Hu, W. W., Taylor, N. F., Lambe, A., Cerully, K., Bougiatioti, A., Liu, P., Krejci, R., Petaja, T., Percival, C., Davidovits, P., Worsnop, D. R., Ekman, A. M. L., Nenes, A., Martin, S., Jimenez, J. L., Collins, D. R., Topping, D. O., Bertram, A. K., Zuend, A., Virtanen, A., and Riipinen, I.: Microphysical explanation of the RH-dependent water affinity of biogenic organic aerosol and its importance for climate, Geophys. Res. Lett., 44, 5167-5177, 2017.

Renbaum-wolff, L., Song, M., Marcolli, C., Zhang, Y., and Liu, P. F.: Observations and implications of liquid − liquid phase separation at high relative humidities in secondary organic material produced by α -pinene ozonolysis without inorganic salts, Atmos. Chem. Phys., 16, 7969-7979, 10.5194/acp-16-7969-2016, 2016.

---

## Author Response (AR2)

Jason Surratt,

Co-Editor of ACP

Dear Jason Surratt,

We thank the reviewer for carefully reading our manuscript and for their very helpful suggestions. Listed below are our responses to the two comments from the reviewer. For clarity and visual distinction, the referee comments are listed here in black and are preceded by bracketed, italicized numbers (e.g. *[1]*). Authors' responses are in red below each referee statement with matching numbers (e.g. *[A1]*).

Sincerely,

Mijung Song
Assistant Professor of Earth and Environmental Sciences
Chonbuk National University

**Referee #1:**

Two comments have not been addressed from reviewer 1:

*[1]* The result from Zhang et al. 2007 and Jimenez et al. 2009 are misstated here. These studies show that 20 – 80% of the mass of non-refractory aerosol particles are organic compounds. This is an average assessment over all particles. So the question about the relative importance of systems that are purely organic compounds is not addressed.

*[A1]* Thank you for your comment. To address the reviewer's comment, we have revised the sentence in the revised manuscript (pg 2, line 3-4).

*[2]* A7: This still seems like a random list of studies about LLPS. These are not all of the studies that show core-shell or phase separated morphologies. Are they simply examples of the type of particles you observe rather than an exhaustive list? If so, that should be noted.

*[A2]* We agree the Referee's comment. To avoid confusion we have deleted the sentences since they are not all of the studies that showed a core-shell morphology in organic particles free of inorganic salt.